# Prenatal Breastfeeding Intention Is Consistently Associated with Breastfeeding Duration Among WIC-Participating Women

**DOI:** 10.3390/nu16244289

**Published:** 2024-12-12

**Authors:** Christopher E. Anderson, Fu-Chi Yang, Shannon E. Whaley

**Affiliations:** 1Division of Research and Evaluation, Public Health Foundation Enterprises (PHFE) WIC, a Program of Heluna Health, City of Industry, CA 91746, USA; 2Department of Biostatistics, University of California Los Angeles, Los Angeles, CA 90095, USA

**Keywords:** breastfeeding duration, breastfeeding intention, WIC, infant feeding, breastfeeding support

## Abstract

Background/Objective: The Special Supplemental Nutrition Program for Women, Infants, and Children (WIC) provides breastfeeding support to participating women in low-income households. This study aimed to determine the relationships between prenatal maternal and household characteristics and breastfeeding duration, as well as whether these characteristics modify associations of prenatal breastfeeding intention with breastfeeding duration. Methods: This is a prospective cohort study of pregnant respondents to Los Angeles County (LAC), California, WIC surveys conducted between 2005 and 2020 (n = 1014). Associations of prenatal breastfeeding intention with duration (months) of any or fully breastfeeding, determined by WIC infant food package issuance, were assessed with linear regression models. Results: Most women reported the intention to breastfeed (67.7%) and perceived breastfeeding support during pregnancy from WIC and family/friends was associated with breastfeeding intention (both *p*-values < 0.0001). Stronger breastfeeding intention, lower maternal BMI, greater maternal age, greater maternal education, paternal cohabitation and employment, and greater breastfeeding support from family/friends were associated with longer duration of any or fully breastfeeding in multivariable models. Stronger breastfeeding intention was more strongly associated with longer duration of any breastfeeding among women with lower BMI (interaction *p*-value 0.03). Conclusions: Breastfeeding support from WIC is an important contributor to stronger breastfeeding intention. Given the robust association of breastfeeding intention with breastfeeding duration, regardless of maternal and household characteristics, WIC breastfeeding support during pregnancy represents an important mechanism to improve breastfeeding outcomes in this population. Further research is needed to understand the directionality of associations between breastfeeding support and intention among WIC participants.

## 1. Introduction

Breastfeeding is an important health-promoting behavior, with numerous reported health benefits for breastfeeding women and their children, including lower risk of type 2 diabetes, obesity, and asthma among children and lower risk of type 2 diabetes, breast cancer, and ovarian cancer among women [1]. The 2020–2025 Dietary Guidelines for Americans and the American Academy of Pediatrics recommend that infants be breastfed exclusively for 6 months, and that breastfeeding be continued following the introduction of complementary foods until at least 1 or 2 years of age [2,3], with breastfeeding considered nutritionally optimal for child development, with appropriate complementary supplementation [4]. Breastfeeding rates in the United States remained far from Healthy People 2030 goals before the COVID-19 pandemic, with only 24.9% of children being breastfed at 6-months of age in 2019 (stable from 2015) [5]. The lower breastfeeding rates reported following the onset of the COVID-19 pandemic may have moved breastfeeding rates further from public health goals [6].

The Special Supplemental Nutrition Program for Women, Infants, and Children (WIC) is a federal nutrition assistance program that provides supplemental food packages, breastfeeding support, nutrition education, and health and social service referrals to pregnant and postpartum women, as well as infants and children up to 5 years of age who live in low-income households [7,8]. To ensure adequate nutrition for the one-half of infants born in the US who are served by WIC [9], when the mother reports not fully breastfeeding, the program issues infant formula to complement the amount of breastfeeding reported [10]. Despite WIC’s breastfeeding support efforts, more than half of participating infants receive infant formula from WIC by the age of 2 months [11]. A systematic review of the association between WIC program participation and breastfeeding outcomes found no association between WIC participation and breastfeeding initiation, that the 2009 food package revisions may have led to increases in breastfeeding exclusivity, and that WIC breastfeeding support may contribute to higher rates of breastfeeding initiation and duration [12].

Associations between prenatal breastfeeding intention and breastfeeding initiation and duration have been reported previously, with definite intention to breastfeed being associated with 24 times higher odds of initiation and 7 times higher odds of breastfeeding for >4 weeks compared to tentative intention [13]. Given the strong association between prenatal breastfeeding intention and breastfeeding outcomes and the suboptimal rates of breastfeeding among WIC-participating households, identifying additional factors that contribute to differences in breastfeeding duration is necessary to inform WIC breastfeeding promotion and support efforts. This study was conducted with the aims of (1) identifying associations between prenatal maternal and household factors and breastfeeding duration and (2) identifying whether the association between breastfeeding intention and duration is modified by maternal and household characteristics.

## 2. Methods

### 2.1. Setting and Sample

This study includes data from 1014 pregnant women living in WIC-participating households who responded during triennial Los Angeles County (LAC) WIC surveys administered in 2005 (n = 183), 2008 (n = 153), 2011 (n = 128), 2014 (n = 154), 2017 (n = 62), and 2020 (n = 334) [14], and who subsequently enrolled an infant in WIC. One infant was selected at random for women who enrolled multiple infants. Response rates were 66, 60, 54, 50, 52, and 53% for 2005, 2008, 2011, 2014, 2017, and 2020 surveys, respectively. All respondents received a USD 10 incentive for completing the survey. Surveys were approved by IRBs at the California Health and Human Services Agency (2017 and 2020), Independent Review Consulting, Inc., and Ethical and Independent Review Services (2005–2014). Oral informed consent was obtained from all study participants for each interview.

### 2.2. Exposures

The primary exposure of interest in this analysis is prenatal breastfeeding intention (know you will breastfeed, think you might breastfeed, know you will not breastfeed, you do not know what to do about breastfeeding). Other variables assessed include survey year (dichotomized: 2020 vs. prior years) and gestational age (months) at the time of survey completion; maternal race/ethnicity-language preference (Asian, Black, English-speaking Hispanic, Spanish-speaking Hispanic, White, Other); age; pre-pregnancy body mass index (weight (kg)/height (m)^2^) from self-reported weight and height (underweight: <18.5 kg/m^2^, normal weight: 18.5 to <25 kg/m^2^, overweight: 25 to <30 kg/m^2^, and obese: ≥30 kg/m^2^); educational attainment (<high school degree, high school degree, >high school degree); depressive symptoms (yes, no); employment (not working for pay, working part time, working full time); paternal cohabitation (yes, no) and employment (not working for pay, working part time, working full time); household food insecurity (yes, no) and Supplemental Nutrition Assistance Program (SNAP) participation (yes, no); pregnancy experiences, including perception of breastfeeding support from WIC (a lot, some, a little, none), perception of breastfeeding support from family or friends (a lot, some, a little, none), perception of difficulty of pregnancy (mostly or very difficult, occasionally difficult or very/mostly easy), perception of pregnancy weight gain to date (just right, too little/much); receipt of gestational diabetes screening (yes, no); gestational age at the start of prenatal care (months); and the number of prenatal care providers.

### 2.3. Breastfeeding Duration

The WIC program issues four infant food packages that vary in the amount of infant formula provided: fully breastfeeding (0 mL of infant formula), mostly breastfeeding (≤5323 mL monthly), some breastfeeding (6624 to 11,918 mL monthly), and no breastfeeding (9927 to 13,071 mL monthly) [10]. The primary outcomes for this analysis include the duration of fully breastfeeding and the duration of any breastfeeding, determined as the number of months of fully breastfeeding package issuance (fully breastfeeding duration) or the number of months of fully, mostly, or some breastfeeding package issuance (any breastfeeding duration) from 0 to 12 months of age.

### 2.4. Statistical Analysis

Survey respondent characteristics were summarized in prenatal breastfeeding intention categories with frequencies and proportions for categorical variables or means and standard deviations for continuous variables. Statistical comparisons for the distribution of respondent characteristics were made with chi-square tests of independence for categorical variables and analysis of variance F tests for continuous variables.

To determine which factors remained significantly associated with breastfeeding intention once mutually adjusted, an ordinal logistic regression model with a cumulative logit link function was performed for breastfeeding intention (dependent variables: categories ordered “know you will breastfeed”, “think you might breastfeed”, “you don’t know what to do about breastfeeding”, “know you will not breastfeed”) based on terms for independent variables, including survey year (2020 or prior years), maternal race/ethnicity -language preference, maternal age (continuous), maternal educational attainment, maternal depressive symptoms during pregnancy, maternal employment status during pregnancy, paternal cohabitation and employment, food insecurity, perceived breastfeeding support by WIC, and perceived breastfeeding support by family/friends. The *p*-values for type 3 score statistics from this ordinal logistic regression model were significant (<0.05) only for survey year, perceived breastfeeding support by WIC, and perceived breastfeeding support by family/friends. Covariates for regression models were selected either *a priori*, based upon reported associations with breastfeeding duration or based upon an identified significant bivariate association with breastfeeding intention in this study. Linear regression models were conducted separately to analyze the duration of any breastfeeding and fully breastfeeding. The regression models included terms for the independent variables, including breastfeeding intention, survey year (2020 or prior years), maternal race/ethnicity-language preference, maternal age (continuous), maternal educational attainment, maternal depressive symptoms during pregnancy, maternal employment status during pregnancy, paternal cohabitation and employment, food insecurity, perceived breastfeeding support by WIC, and perceived breastfeeding support by family/friends. Due to substantial missing data for the maternal pre-pregnancy BMI category, a second set of models incorporating this variable were conducted. The modification of the association between breastfeeding intention and breastfeeding duration by factors available to local agency WIC programs was evaluated using linear regression models, identically parameterized to those previously described, that incorporated two-way interaction terms between breastfeeding intention and one potential effect modifier (survey year, maternal race/ethnicity-language preference, maternal educational attainment, maternal pre-pregnancy BMI category, household food insecurity, household SNAP participation). The models evaluating effect modification by BMI category and SNAP participation included an independent term for the BMI category and SNAP participation, respectively. All analyses were conducted with SAS 9.4 (SAS Institute Inc., Cary, NC, USA, 2014). All *p*-values <0.05 were considered statistically significant.

## 3. Results

There were 1014 pregnant respondents to surveys from 2005 to 2020. Over two-thirds of respondents indicated they know they will breastfeed (67.7%), with fewer stating they think they might breastfeed (19.3%), they know they will not breastfeed (5.1%), or are unsure of what to do about breastfeeding (7.9%) (Table 1). Maternal educational attainment differed by breastfeeding intention, with a higher proportion of know they will (35.5%) or think they might breastfeed (34.7%) respondents having greater education than high school completion compared to the respondents who know they will not (31%) or are unsure (22%) about breastfeeding (*p*-value 0.05). Women who know they will (70.0%) or think they might (69.9%) breastfeed were more likely to be cohabiting with the infant’s father than women who know they will not breastfeed (52%) (*p*-value = 0.02). The cohabiting fathers of the children of women who know they will or think they might breastfeed were more likely to be employed (80.0% and 85.4%, respectively) than those of the children of women who know they will not breastfeed (63%) (*p*-value = 0.01). The women who know they will or think they might breastfeed were less likely to live in a SNAP-participating household (35.3% and 42.4%, respectively) compared to women who know they will not breastfeed (61%) (*p*-value = 0.02). Women who know they will or think they might breastfeed were more likely to perceive a lot of breastfeeding support compared to women who know they will not breastfeed from both WIC (75.4%, 66.3%, and 45%, respectively) and family/friends (66.6%, 50.0%, and 27%, respectively) (both *p*-values < 0.0001). Breastfeeding duration decreased with lower certainty of intention to breastfeed, ranging from know they will, to think they might, to are unsure what to do, and to know they will not. This was observed for the duration of fully breastfeeding (3.0, 1.2, 0.3, and 0.4 months, respectively) and any breastfeeding (6.8, 4.2, 1.2, and 2.9 months, respectively) (both *p*-values < 0.0001). The no breastfeeding package was received by 273 (26.9%) infants in their first month of life, giving them a breastfeeding duration value of 0 months. This proportion increased with lower certainty of breastfeeding intention (*p*-value < 0.0001). The respondents to the 2020 survey accounted for a greater proportion of women who know they will not breastfeed (46%) than of women who know they will (30.8%), think they might (35.7%), or are unsure about what to do about breastfeeding (36%), though this did not achieve statistical significance. Among the 2020 survey respondents, the COVID-19 pandemic changed breastfeeding intention for a larger proportion of mothers who were uncertain about breastfeeding (34%) than for those who either knew they would (13.9%), thought they might (10.3%), or knew they would not (13%) breastfeed (*p*-value = 0.02).

Table 2 presents associations between maternal and household factors with breastfeeding duration. Due to the similarity in results between the models adjusted for maternal pre-pregnancy BMI category and those unadjusted, only the results from the BMI-adjusted models will be discussed. Thinking you might breastfeed was associated with shorter durations of any (−2.24 months, 95% CI [−3.13, −1.36]) and fully (−1.60 months [−2.24, −0.96]) breastfeeding. Knowing you will not breastfeed was associated with shorter durations of any (−5.01 [−6.34, −3.68]) and fully (−2.50 months [−3.30, −1.70]) breastfeeding, and being unsure about breastfeeding was associated with shorter durations of any (−3.72 months [−5.02, −2.42]) and fully (−2.77 months [−3.42, −2.12]) breastfeeding compared to knowing you will breastfeed. Responding to the 2020 survey was associated with shorter duration of any breastfeeding (−1.06 months [−1.81, −0.31]). Black (−2.20 months [−3.51, −0.90]) and English-speaking Hispanic (−1.00 months [−1.77, −0.22]) respondents had shorter duration of any breastfeeding, and Asian respondents had shorter duration of fully breastfeeding (−1.50 months [−2.74, −0.27]), compared to Spanish-speaking Hispanic respondents. Each 5-year increase in maternal age was associated with longer duration (0.64 months [0.34, 0.93]) of any breastfeeding. Maternal pre-pregnancy obesity, compared to normal BMI, was associated with shorter duration (−1.27 months [−2.13, −0.41]) of any breastfeeding. Maternal educational attainment of less than high school completion was associated with shorter duration of fully breastfeeding (−0.73 months [−1.42, −0.04]) compared to greater than high school completion, and maternal depressive symptoms during pregnancy were associated with shorter duration of fully breastfeeding (−0.90 months [−1.68, −0.12]). A father cohabiting and being employed part time was associated with longer duration of any (1.19 months [0.07, 2.31]) and fully (1.34 months [0.32, 2.36]) breastfeeding compared to mothers who were not cohabiting. A father cohabiting and being employed full time was associated with longer duration of any breastfeeding (1.12 months [0.05, 2.19]) compared to mothers who were not cohabiting. A woman reporting some breastfeeding support from family/friends during pregnancy was associated with longer duration of fully breastfeeding (1.27 months [0.15, 2.39]) compared to women who reported no breastfeeding support from family/friends.

The association between breastfeeding intention and fully breastfeeding duration was not modified by any of the examined potential effect modifiers (Table 3). The magnitude of association between thinking you might or knowing you will not breastfeed compared to knowing you will breastfeed and any breastfeeding duration was smaller for women with overweight pre-pregnancy BMI (−1.73 months [−3.49, 0.03] and −6.01 months [−7.11, −4.90] for think you might and know you will not, respectively) and obese pre-pregnancy BMI (−1.92 months [−3.31, −0.53] and -2.73 months [−5.21, −0.25] for think you might and know you will not, respectively) compared to women with normal weight pre-pregnancy BMI (−2.93 months [−4.38, −1.48] and −7.14 months [−8.29, −5.99] for think you might and know you will not, respectively) (interaction *p*-value = 0.03).

## 4. Discussion

In this study of breastfeeding outcomes among a sample of pregnant WIC-participating women in LAC, California, it was found that the majority of women intended to breastfeed. The factors associated with stronger breastfeeding intention included greater maternal education, paternal cohabitation and employment, and greater perceived breastfeeding support from WIC and family/friends. In multivariable models, breastfeeding intention, maternal race/ethnicity, age, pre-pregnancy BMI, educational attainment, depression during pregnancy, paternal cohabitation and employment, and breastfeeding support from family/friends were associated with the duration of any breastfeeding or fully breastfeeding. Pre-pregnancy BMI was the only factor that modified the association between breastfeeding intention and the duration of any or fully breastfeeding.

Indicators of lower household socioeconomic status, including lower maternal educational attainment, a non-cohabiting or cohabiting but unemployed father, and SNAP participation were more prevalent among pregnant women with lower intention to breastfeed. These findings align with prior results, which found that the intention to breastfeed is stronger among women with higher education [15,16,17] and married women or women cohabiting with the child’s father [17,18,19]. While we are unaware of any prior studies finding an association between paternal employment during pregnancy and maternal breastfeeding intention, higher educational attainment is associated with both lower rates of unemployment [20] and higher rates of cohabitation or marriage [21] and of breastfeeding [22] and could contribute to the observed association of greater paternal employment with stronger maternal breastfeeding intention.

Perceived support of breastfeeding from both WIC and family or friends were strongly associated with breastfeeding intention. An association between perceived support of a partner and intention to continue breastfeeding among employed women has been previously reported [23]. Following adjustment for prenatal breastfeeding intention, the present study identified a significant association between greater perceived family/friend support of breastfeeding and longer duration of fully breastfeeding, but did not identify an association between perceived breastfeeding support from WIC and duration of any or fully breastfeeding. A number of recent publications have identified significant associations between WIC and maternity care breastfeeding support policies and breastfeeding outcomes [24,25]; however, these did not adjust for intention to breastfeed [26]. Based on these patterns of association, it is hypothesized that stronger breastfeeding intention mediates the association between perceived WIC breastfeeding support and breastfeeding outcomes.

A stronger breastfeeding intention was associated with a longer duration of both any breastfeeding and fully breastfeeding in this study, aligning with prior literature, which found that breastfeeding intention is strongly associated with breastfeeding practices early in infancy [13]. This study also identified significantly shorter duration of any breastfeeding among Black and English-speaking Hispanic women and significantly shorter duration of fully breastfeeding among Asian women, compared to Spanish-speaking Hispanic women, aligning with prior reported racial/ethnic disparities in breastfeeding [27]. Greater maternal age and greater maternal educational attainment were associated with longer duration of any breastfeeding and fully breastfeeding, respectively, aligning with prior studies [27]. The association between depression during pregnancy and shortened duration of fully breastfeeding was expected, given a previously reported association between prenatal depression and breastfeeding duration, but not breastfeeding intention [28]. This study identified an association between living with the infant’s employed father during pregnancy and longer breastfeeding duration, aligning with prior findings that paternal employment is associated with lower odds of never breastfeeding among pregnant women in California [29].

The duration of any breastfeeding was shorter among women who were pregnant in 2020 compared to 2005–2017, and this was robust to adjustment for prenatal breastfeeding intention, though, unexpectedly, the survey year did not significantly modify the association between breastfeeding intention and breastfeeding duration. The shorter breastfeeding durations for infants born to women pregnant during 2020 align with prior results reporting lower rates of breastfeeding during the COVID-19 pandemic that coincide with departures from recommended practices to promote breastfeeding immediately after birth [30]. A prior study of a representative sample of WIC-participating infants in LAC found that among infants born in 2018–2020, those born before March 2020 were significantly more likely to be fully breastfeeding at 1-, 3-, and 6-months of age and significantly more likely to be receiving any breastfeeding at 3- and 6-months of age compared to those born during or after March 2020 [6]. The present study only identified an association between pregnancy in 2020 and a shorter duration of any breastfeeding, but not with duration of fully breastfeeding, in contrast to the prior results [6]. However, the present results compared children born to women who were pregnant in 2005–2017 to those who were pregnant in 2020. In contrast, a study of births in Wales from 2018 to 2021 identified higher exclusive breastfeeding rates at age 6 months for infants born in 2020 and lower rates for infants born in 2021 compared to the births in 2018 [31]. The present study did not identify an association between survey year and fully breastfeeding duration, potentially due to low rates of fully breastfeeding among WIC participants [11,32,33].

Maternal pre-pregnancy BMI was not significantly associated with breastfeeding intention, in contrast to findings from a systematic review that reported a lower intention to breastfeed among women with obesity [34]. This difference may be attributable to the universal WIC participation in the present study, with high perceived breastfeeding support from WIC being reported (82% of participants reported receiving a lot or some breastfeeding support from WIC). Any breastfeeding duration among women with pre-pregnancy obesity was significantly shorter compared to women with a healthy BMI, aligning with much of the prior literature that found lower breastfeeding rates among women with higher pre-pregnancy BMI [34,35,36]. The association between breastfeeding intention and duration was significantly stronger among women with healthy pre-pregnancy BMI compared to those who were overweight or obese, suggestive of greater difficulty meeting breastfeeding intention among women with pre-pregnancy overweight and obesity, and aligning with findings of prior studies [34,35,36]. Lower breastmilk transfer [37] and perceived insufficient milk supply [38] have been reported among women with obesity, which might explain both the shorter duration of any breastfeeding among women with pre-pregnancy obesity in the present study and the weaker association between breastfeeding intention and duration of any breastfeeding among women with pre-pregnancy overweight or obesity in the present study.

The strengths of this study include the prenatal assessment of breastfeeding intention and the prospectively collected outcome data (WIC infant food package), which have been previously validated as a proxy for infant feeding practices [39]. The study participants were representative of pregnant women in WIC-participating households in LAC, as they were randomly selected for participation in the triennial LAC WIC Survey. The limitations include the cross-sectional assessment of all covariates, including effect modifiers, precluding the assessment of directionality of associations in the baseline data (e.g., if women with greater intent to breastfeed are more likely to perceive greater levels of support of breastfeeding from WIC and family, or whether greater levels of support of breastfeeding from WIC and family contribute to greater intent to breastfeed). Care should be taken when generalizing the results of this study to populations of different demographics, as the study participants were all WIC participants, primarily Hispanic, and lived in low-income households, in a large urban county in Southern California. Given the strength of the relationship between perceived WIC breastfeeding support during pregnancy and breastfeeding intention, the relationships between breastfeeding intention and duration among women not participating in WIC during pregnancy may be of different magnitude.

## 5. Conclusions

In conclusion, most WIC-participating pregnant women in LAC, California, intend to breastfeed their infant. This study identified that women with a stronger intention to breastfeed perceived stronger support of breastfeeding from WIC and family/friends during their pregnancy. However, after adjusting for breastfeeding intention, perceived support of breastfeeding from WIC was not associated with breastfeeding duration. Breastfeeding intention was robustly associated with longer duration of any and fully breastfeeding and was not modified by socioeconomic or demographic characteristics of the women or their households. These results indicate that WIC breastfeeding support may be an important contributor to a greater breastfeeding intention, and that breastfeeding intention is strongly associated with breastfeeding duration regardless of maternal and household characteristics. Further research is needed to understand which maternal and household characteristics contribute most to prenatal breastfeeding intention to inform future programmatic efforts to increase breastfeeding intention, and to understand the directionality of associations between breastfeeding support and intention in this population. WIC breastfeeding support during pregnancy may represent an important mechanism for improving breastfeeding outcomes in this population.

## Figures and Tables

**Table 1 nutrients-16-04289-t001:** Characteristics of WIC-participating respondents to the prenatal version of LA County WIC Survey between 2005 and 2020, categorized by breastfeeding intention (n = 1014).

		Will Breastfeed	Might Breastfeed	Will Not Breastfeed	Uncertain About Breastfeeding	
**Variable**		n = 686	n = 196	n = 52	n = 80	*p*-value
Survey year, n (%)						0.08
	2020	211 (30.8)	70 (35.7)	24 (46)	29 (36)	
	2005–2017	475 (69.2)	126 (64.3)	28 (54)	51 (64)	
Pandemic changed breastfeeding intention, n (%) ^a^		29 (13.9)	7 (10.3)	3 (13)	10 (34)	0.02
**Maternal and household characteristics:**						
Maternal race/ethnicity-language, n (%)						0.32
	Asian	38 (5.5)	12 (6.1)	2 (4)	1 (1)	
	Black	47 (6.9)	17 (8.7)	3 (6)	2 (3)	
	Hispanic, EN	227 (33.1)	48 (24.5)	21 (40)	27 (34)	
	Hispanic, SP	344 (50.1)	113 (57.7)	23 (44)	48 (60)	
	White	24 (3.5)	5 (2.6)	2 (4)	1 (1)	
	Other	6 (0.9)	1 (0.5)	1 (2)	1 (1)	
Maternal age (years), n (%)						0.99
	<25 years	209 (30.5)	63 (32.1)	15 (29)	25 (31)	
	25 to <30 years	186 (27.1)	52 (26.5)	16 (31)	23 (29)	
	30 to <35 years	166 (24.2)	50 (25.5)	12 (23)	18 (23)	
	≥35 years	125 (18.2)	31 (15.8)	9 (17)	14 (18)	
Maternal educational attainment, n (%)						0.05
	<High school completed	245 (36.1)	70 (35.7)	20 (38)	44 (56)	
	Completed high school	193 (28.4)	58 (29.6)	16 (31)	18 (23)	
	>High school completed	241 (35.5)	68 (34.7)	16 (31)	17 (22)	
Mother screened positive for depression, n (%)		95 (14.0)	23 (11.8)	5 (10)	14 (18)	0.53
Mother not employed, n (%)		490 (71.4)	146 (74.5)	38 (73)	65 (81)	0.28
Father living in household, n (%)		480 (70.0)	137 (69.9)	27 (52)	62 (78)	0.02
Father not employed, n (%) ^b^		96 (20.0)	20 (14.6)	10 (37)	8 (13)	0.01
Food insecure, n (%)		207 (30.2)	67 (34.2)	12 (23)	29 (36)	0.30
SNAP participation, n (%) ^c^		164 (35.3)	56 (42.4)	20 (61)	21 (44)	0.02
**Pregnancy experiences:**						
Breastfeeding support from WIC, n (%)						<0.0001
	A lot	510 (75.4)	128 (66.3)	23 (45)	35 (45)	
	Some	70 (10.4)	31 (16.1)	14 (27)	17 (22)	
	A little	50 (7.4)	14 (7.3)	6 (12)	12 (15)	
	None	46 (6.8)	20 (10.4)	8 (16)	14 (18)	
Breastfeeding support from family/friends, n (%)						<0.0001
	A lot	455 (66.6)	98 (50.0)	14 (27)	41 (51)	
	Some	127 (18.6)	63 (32.1)	19 (37)	19 (24)	
	A little	53 (7.8)	23 (11.7)	4 (8)	9 (11)	
	None	48 (7.0)	12 (6.1)	15 (29)	11 (14)	
Pregnancy has been mostly or very difficult, n (%)		64 (9.4)	22 (11.6)	8 (15)	12 (15)	0.23
Feel pregnancy weight gain just right, n (%)		524 (79.3)	135 (69.9)	38 (75)	60 (79)	0.05
Gestational diabetes screening, n (%) ^d^		191 (54.1)	60 (57.1)	18 (62)	16 (43)	0.41
Pre-pregnancy weight (pounds), mean ± SD		155 ± 39	159 ± 41	164 ± 38	155 ± 39	0.24
Pre-pregnancy BMI (kg/m^2^),^e^ mean ± SD		27.7 ± 6.2	28.8 ± 6.8	28.9 ± 5.7	28.3 ± 6.6	0.19
Month at survey, mean ± SD		7.0 ± 1.9	7.1 ± 1.9	7.3 ± 1.7	6.7 ± 1.9	0.28
Month started prenatal care, mean ± SD		2.0 ± 1.4	2.0 ± 1.5	1.9 ± 0.9	2.1 ± 1.5	0.77
Number of prenatal care providers, ^d^ mean ± SD		1.5 ± 1.0	1.6 ± 1.2	1.3 ± 0.6	1.5 ± 1.1	0.15
**Infant feeding:**						
Zero months of any breastfeeding, n (%)		113 (16.5)	76 (38.8)	40 (76.9)	44 (55.0)	<0.0001
Months of fully breastfeeding, mean ± SD		3.0 ± 4.7	1.2 ± 3.2	0.3 ± 1.4	0.4 ± 1.7	<0.0001
Months of any breastfeeding, mean ± SD		6.8 ± 5.3	4.2 ± 5.1	1.2 ± 3.2	2.9 ± 4.7	<0.0001

Abbreviations: BMI, body mass index; EN, English-speaking; LA, Los Angeles; SD, standard deviation; SNAP, Supplemental Nutrition Assistance program; SP, Spanish-speaking; WIC, Special Supplemental Nutrition Program for Women, Infants, and Children. ^a^ The impact of the pandemic on breastfeeding intention was only assessed among survey respondents in 2020 (n = 334). ^b^ The employment of the child’s father was only assessed if the father was living in the household with the mother (n = 706). ^c^ SNAP participation was only assessed in surveys from 2011 to 2020 (n = 677). ^d^ The denominator for prenatal care variables includes only those women who reported having begun prenatal care (n = 965). For gestational diabetes screening, only respondents from 2014 to 2020 were included (n = 524). ^e^ The maternal pre-pregnancy BMI was calculated for women who reported both height and weight (n = 840).

**Table 2 nutrients-16-04289-t002:** Association between prenatal maternal and household characteristics and duration of fully and any breastfeeding among WIC-participating women who completed an LA County WIC Survey, 2005–2020 (n = 1014).

		Any Breastfeeding Duration (Months)		Fully Breastfeeding Duration (Months)	
Variable		Est (95% CI) ^a^	Est (95% CI) ^b^	Est (95% CI) ^a^	Est (95% CI) ^b^
In regard to breastfeeding, you…					
	Know you will breastfeed	0.00 (ref)	0.00 (ref)	0.00 (ref)	0.00 (ref)
	Think you might breastfeed	−2.49 (−3.28, −1.69)	−2.24 (−3.13, −1.36)	−1.77 (−2.34, −1.20)	−1.60 (−2.24, −0.96)
	Know you will not breastfeed	−5.27 (−6.43, −4.11)	−5.01 (−6.34, −3.68)	−2.65 (−3.35, −1.95)	−2.50 (−3.30, −1.70)
	Are unsure about breastfeeding	−4.07 (−5.21, −2.94)	−3.72 (−5.02, −2.42)	−2.68 (−3.25, −2.11)	−2.77 (−3.42, −2.12)
Survey in 2020		−0.84 (−1.53, −0.14)	−1.06 (−1.81, −0.31)	0.11 (−0.47, 0.69)	0.03 (−0.60, 0.66)
Maternal race/ethnicity-language preference					
	Asian	0.56 (−0.92, 2.04)	0.15 (−1.42, 1.72)	−0.92 (−2.14, 0.30)	−1.50 (−2.74, −0.27)
	Black	−2.21 (−3.45, −0.97)	−2.20 (−3.51, −0.90)	−0.74 (−1.71, 0.22)	−0.61 (−1.64, 0.43)
	Hispanic, EN	−0.97 (−1.68, −0.25)	−1.00 (−1.77, −0.22)	−0.49 (−1.08, 0.10)	−0.51 (−1.16, 0.14)
	Hispanic, SP	0.00 (ref)	0.00 (ref)	0.00 (ref)	0.00 (ref)
	White	−1.56 (−3.37, 0.26)	−1.16 (−3.13, 0.80)	0.37 (−1.35, 2.08)	0.57 (−1.35, 2.49)
	Other	1.57 (−1.86, 5.01)	1.62 (−1.74, 4.98)	2.36 (−0.56, 5.28)	2.31 (−0.45, 5.08)
Maternal age (5 year increase)		0.46 (0.20, 0.73)	0.64 (0.34, 0.93)	0.16 (−0.05, 0.38)	0.23 (−0.01, 0.47)
Maternal pre-pregnancy BMI					
	Underweight	-	0.16 (−2.19, 2.51)	-	−0.05 (−2.15, 2.06)
	Normal weight	-	0.00 (ref)	-	0.00 (ref)
	Overweight	-	−0.43 (−1.29, 0.44)	-	−0.29 (−1.01, 0.43)
	Obese	-	−1.27 (−2.13, −0.41)	-	−0.51 (−1.20, 0.18)
Maternal educational attainment					
	<High school completed	0.28 (−0.55, 1.10)	0.25 (−0.64, 1.14)	−0.71 (−1.36, −0.07)	−0.73 (−1.42, −0.04)
	Completed high school	−0.45 (−1.26, 0.36)	−0.28 (−1.12, 0.56)	−0.58 (−1.25, 0.09)	−0.67 (−1.38, 0.03)
	>High school completed	0.00 (ref)	0.00 (ref)	0.00 (ref)	0.00 (ref)
Depressed during pregnancy		−0.49 (−1.40, 0.42)	−0.55 (−1.59, 0.49)	−0.41 (−1.13, 0.31)	−0.90 (−1.68, −0.12)
Maternal employment					
	Not employed	0.00 (ref)	0.00 (ref)	0.00 (ref)	0.00 (ref)
	Employed part time	0.90 (−0.11, 1.91)	0.85 (−0.22, 1.92)	−0.42 (−1.20, 0.35)	−0.25 (−1.09, 0.59)
	Employed full time	0.29 (−0.66, 1.23)	0.24 (−0.78, 1.25)	−0.31 (−1.07, 0.46)	−0.26 (−1.09, 0.58)
Father is…					
	Not cohabiting	0.00 (ref)	0.00 (ref)	0.00 (ref)	0.00 (ref)
	Cohabiting, not employed	−0.27 (−1.06, 0.51)	−0.14 (−0.99, 0.71)	−0.46 (−1.07, 0.15)	−0.15 (−0.81, 0.50)
	Cohabiting, employed part time	0.76 (−0.28, 1.81)	1.19 (0.07, 2.31)	1.00 (0.04, 1.95)	1.34 (0.32, 2.36)
	Cohabiting, employed full time	1.26 (0.28, 2.23)	1.12 (0.05, 2.19)	0.45 (−0.34, 1.23)	0.69 (−0.20, 1.58)
Food insecure		0.15 (−0.57, 0.88)	0.08 (−0.70, 0.86)	−0.04 (−0.62, 0.54)	0.22 (−0.43, 0.87)
Breastfeeding support from WIC					
	A lot	−0.35 (−1.44, 0.74)	−0.11 (−1.34, 1.12)	−0.33 (−1.22, 0.57)	−0.65 (−1.70, 0.39)
	Some	−0.06 (−1.34, 1.22)	−0.31 (−1.70, 1.09)	−0.27 (−1.32, 0.78)	−0.84 (−2.00, 0.32)
	A little	0.89 (−0.57, 2.36)	1.25 (−0.37, 2.86)	0.51 (−0.66, 1.68)	0.29 (−1.06, 1.64)
	None	0.00 (ref)	0.00 (ref)	0.00 (ref)	0.00 (ref)
Breastfeeding support from family/friends					
	A lot	0.25 (−0.99, 1.48)	−0.37 (−1.76, 1.03)	0.90 (0.05, 1.75)	0.99 (−0.04, 2.02)
	Some	0.07 (−1.24, 1.38)	−0.44 (−1.90, 1.02)	1.32 (0.35, 2.29)	1.27 (0.15, 2.39)
	A little	−0.26 (−1.76, 1.25)	−0.96 (−2.64, 0.72)	0.49 (−0.53, 1.51)	0.43 (−0.76, 1.63)
	None	0.00 (ref)	0.00 (ref)	0.00 (ref)	0.00 (ref)

Abbreviations: BMI, body mass index; EN, English-speaking; LA, Los Angeles; SP, Spanish-speaking; WIC, Special Supplemental Nutrition Program for Women, Infants, and Children; ^a^ Estimates (95% CI) represent the difference in the duration of any breastfeeding or fully breastfeeding associated with each of the specified independent variables, unadjusted for maternal pre-pregnancy BMI. These estimates were calculated using linear regression models for the number of months of either fully breastfeeding duration (months of issuance of the fully breastfeeding infant package) or any breastfeeding duration (months of issuance of the fully or partly breastfeeding infant packages), with terms included in the regression models for breastfeeding intention, a dichotomous indicator for the survey year (2020 or prior years), maternal race/ethnicity-language preference, maternal age (continuous), maternal educational attainment, maternal depressive symptoms during pregnancy, maternal employment status during pregnancy, paternal cohabitation and employment, food insecurity, perceived breastfeeding support by WIC, and perceived breastfeeding support by family/friends (n = 1014). ^b^ Estimates (95% CI) represent the difference in any breastfeeding duration or fully breastfeeding duration associated with the specified variables, adjusted for maternal pre-pregnancy BMI. These estimates were calculated using linear regression models for the number of months of either fully breastfeeding duration (months of issuance of the fully breastfeeding infant package) or any breastfeeding duration (months of issuance of the fully or partly breastfeeding infant packages), with terms included in the regression models for breastfeeding intention, a dichotomous indicator for the survey year (2020 or prior years), maternal race/ethnicity-language preference, maternal age (continuous), maternal pre-pregnancy BMI category, maternal educational attainment, maternal depressive symptoms during pregnancy, maternal employment status during pregnancy, paternal cohabitation and employment, food insecurity, perceived breastfeeding support by WIC, and perceived breastfeeding support by family/friends (n = 840).

**Table 3 nutrients-16-04289-t003:** Associations of breastfeeding intention with duration of any breastfeeding and fully breastfeeding by levels of potential effect modifiers among WIC-participating women who completed a LA County WIC Survey, 2005–2020 (n = 1014).

	Any Breastfeeding Duration (Months)				Fully Breastfeeding Duration (Months)			
	Think you Might vs. Know You Will	Know You Will Not vs. Know You Will	Unsure vs. Know You Will	*p* ^d^	Think You Might vs. Know You Will	Know You Will Not vs. Know You Will	Unsure vs. Know You Will	*p* ^d^
**Survey year ^a^**				0.53				0.12
2020	−2.29 (−3.64, −0.95)	−6.04 (−7.08, −5.00)	−4.09 (−5.75, −2.42)		−2.15 (−3.15, −1.14)	−3.30 (−4.19, −2.41)	−2.32 (−3.53, −1.12)	
2005–2017	−2.59 (−3.59, −1.60)	−4.65 (−6.47, −2.84)	−4.07 (−5.59, −2.56)		−1.58 (−2.27, −0.88)	−2.16 (−3.11, −1.20)	−2.89 (−3.44, −2.34)	
**Race/ethnicity ^a^**				0.28				0.83
Asian	−1.07 (−4.60, 2.46)	−6.37 (−8.49, −4.25)	−5.19 (−7.13, −3.25)		−1.75 (−4.25, 0.76)	−2.29 (−4.08, −0.51)	−2.60 (−4.26, −0.94)	
Black	−1.22 (−3.32, 0.88)	1.25 (−4.08, 6.59)	−3.91 (−5.64, −2.19)		−1.54 (−3.02, −0.07)	1.26 (−3.48, 6.00)	−3.26 (−5.01, −1.51)	
Hispanic, SP	−3.66 (−5.06, −2.26)	−6.47 (−7.39, −5.55)	−5.41 (−6.56, −4.25)		−1.80 (−2.79, −0.81)	−2.67 (−3.40, −1.95)	−2.63 (−3.35, −1.92)	
Hispanic, EN	−2.09 (−3.23, −0.96)	−4.75 (−6.71, −2.79)	−3.45 (−5.19, −1.71)		−1.68 (−2.50, −0.86)	−2.81 (−3.66, −1.96)	−2.58 (−3.44, −1.72)	
White	−5.70 (−8.01, −3.40)	−7.63 (−10.02, −5.25)	−7.28 (−9.75, −4.81)		−4.80 (−7.12, −2.48)	−5.64 (−7.96, −3.33)	−6.22 (−8.58, −3.87)	
Other	−0.71 (−5.11, 3.70)	−6.13 (−10.48, −1.78)	5.48 (1.27, 9.69)		1.19 (−3.17, 5.55)	−5.22 (−9.51, −0.92)	−3.35 (−7.52, 0.82)	
**Mom education ^a^**				0.76				0.93
<HS completed	−2.71 (−4.08, −1.35)	−5.48 (−7.37, −3.58)	−4.87 (−6.46, −3.29)		−1.56 (−2.43, −0.69)	−2.33 (−3.00, −1.65)	−2.73 (−3.39, −2.06)	
Completed HS	−2.10 (−3.51, −0.69)	−4.46 (−6.30, −2.62)	−3.60 (−5.53, −1.67)		−1.85 (−2.91, −0.79)	−2.75 (−3.82, −1.68)	−3.00 (−3.77, −2.23)	
>HS completed	−2.58 (−3.91, −1.24)	−5.86 (−7.85, −3.88)	−2.83 (−5.32, −0.34)		−1.92 (−2.95, −0.90)	−2.94 (−4.56, −1.32)	−2.16 (−3.86, −0.45)	
**Maternal BMI ^b^**				0.03				0.53
Normal/underweight	−2.93 (−4.38, −1.48)	−7.14 (−8.29, −5.99)	−2.84 (−5.00, −0.67)		−1.66 (−2.69, −0.62)	−3.29 (−4.31, −2.26)	−2.57 (−3.81, −1.33)	
Overweight	−1.73 (−3.49, 0.03)	−6.01 (−7.11, −4.90)	−2.87 (−5.44, −0.30)		−1.38 (−2.69, −0.07)	−2.95 (−3.94, −1.96)	−2.86 (−3.76, −1.96)	
Obese	−1.92 (−3.31, −0.53)	−2.73 (−5.21, −0.25)	−5.28 (−7.20, −3.37)		−1.69 (−2.69, −0.69)	−1.63 (−3.05, −0.21)	−2.92 (−3.84, −1.99)	
**Food insecurity ^a^**				0.73				0.52
Food insecure	−2.23 (−3.73, −0.74)	−4.99 (−7.82, −2.17)	−4.81 (−6.67, −2.96)		−1.72 (−2.76, −0.69)	−2.11 (−3.34, −0.88)	−2.95 (−3.74, −2.15)	
Food secure	−2.61 (−3.55, −1.67)	−5.33 (−6.56, −4.10)	−3.69 (−5.12, −2.26)		−1.80 (−2.48, −1.12)	−2.78 (−3.58, −1.99)	−2.54 (−3.28, −1.80)	
**SNAP ^c^**				0.78				0.74
Yes	−2.70 (−4.22, −1.19)	−6.48 (−7.92, −5.03)	−4.74 (−6.60, −2.89)		−2.31 (−3.45, −1.18)	−3.45 (−4.61, −2.29)	−2.92 (−3.88, −1.96)	
No	−1.73 (−3.04, −0.41)	−5.99 (−7.56, −4.42)	−4.87 (−6.52, −3.22)		−1.56 (−2.57, −0.54)	−3.51 (−4.38, −2.65)	−2.82 (−4.07, −1.57)	

Abbreviations: BMI, body mass index; EN, English-speaking; HS, high school; LA, Los Angeles; SNAP, Supplemental Nutrition Assistance Program; SP, Spanish-speaking; WIC, Special Supplemental Nutrition Program for Women, Infants, and Children; ^a^ Estimates (95% CI) represent the difference in any breastfeeding duration or fully breastfeeding duration associated with breastfeeding intention. These estimates were calculated using linear regression models for the number of months of either fully breastfeeding duration (months of issuance of the fully breastfeeding infant package) or any breastfeeding duration (months of issuance of the fully or partly breastfeeding infant packages), with terms included in the regression models for breastfeeding intention, a dichotomous indicator for the survey year (2020 or prior years), maternal race/ethnicity-language preference, maternal age (continuous), maternal educational attainment, maternal depressive symptoms during pregnancy, maternal employment status during pregnancy, paternal cohabitation and employment, food insecurity, perceived breastfeeding support by WIC, and perceived breastfeeding support by family/friends. The models included a two-way interaction between the specified variable and breastfeeding intention to get stratum-specific estimates for the association between breastfeeding intention and breastfeeding duration (n = 1014). ^b^ Estimates (95% CI) represent the difference in any breastfeeding duration or fully breastfeeding duration associated with breastfeeding intention. These estimates were calculated using linear regression models for the number of months of either fully breastfeeding duration (months of issuance of the fully breastfeeding infant package) or any breastfeeding duration (months of issuance of the fully or partly breastfeeding infant packages), with terms included in the regression models for breastfeeding intention, a dichotomous indicator for the survey year (2020 or prior years), maternal race/ethnicity-language preference, maternal age (continuous), maternal pre-pregnancy BMI category, maternal educational attainment, maternal depressive symptoms during pregnancy, maternal employment status during pregnancy, paternal cohabitation and employment, food insecurity, perceived breastfeeding support by WIC, and perceived breastfeeding support by family/friends. The models included a two-way interaction between the maternal pre-pregnancy BMI category and breastfeeding intention to get stratum-specific estimates for the association between breastfeeding intention and breastfeeding duration (n = 840). ^c^ Estimates (95% CI) represent the difference in any breastfeeding duration or fully breastfeeding duration associated with breastfeeding intention. These estimates were calculated using linear regression models for the number of months of either fully breastfeeding duration (months of issuance of the fully breastfeeding infant package) or any breastfeeding duration (months of issuance of the fully or partly breastfeeding infant packages), with terms included in the regression models for breastfeeding intention, a dichotomous indicator for the survey year (2020 or prior years), maternal race/ethnicity-language preference, maternal age (continuous), household SNAP participation, maternal educational attainment, maternal depressive symptoms during pregnancy, maternal employment status during pregnancy, paternal cohabitation and employment, food insecurity, perceived breastfeeding support by WIC, and perceived breastfeeding support by family/friends. Models included a two-way interaction between household SNAP participation and breastfeeding intention to get stratum-specific estimates for the association between breastfeeding intention and breastfeeding duration (n = 677). ^d^ *p*-values represent the type 3 test of the interaction between breastfeeding intention and the specified potential effect modifier.

## Data Availability

Data used in this study include confidential participant data of the WIC program and will not be made publicly available due to a memorandum of understanding with the California Department of Public Health WIC program.

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
