# Peer review of "Prenatal Breastfeeding Intention Is Consistently Associated with Breastfeeding Duration Among WIC-Participating Women"

_nutrients, 2024, doi:10.3390/nu16244289_

Round 1

Reviewer 1 Report

Comments and Suggestions for Authors

This is a straightforward paper that reports important and useful data that also has a number of significant practical applications in the important area of promoting breast feeding in infancy.

However, there are a number issues I would like the authors to consider before a determination to whether the paper can be accepted for publication.

1.        Title:  It may just be a matter of differing styles of writing but I do not like the word “Robustly” in the title.  I cannot recall seeing the word used any where before to describe the strength of a relationship.  I personally think that the word robustly could be replaced by significant or highly significant and the title wording changed slightly to accommodate the replacement.

2.        The Introduction is well written and provides a good background to the study. I do think however that on Page 2 Line 52, the reader would benefit from a sentence or two that expands a little on the statement

more than half of participating infants receive infant formula from WIC by age 2 months.

The question that needs answering by the authors in my view is Why?

3.        The methods section including the statistical analysis sub section are straightforward and appropriate.

4.        Results are displayed well and easy to interpret. On Page 4 Table 1, whilst it can be easily inferred I think for completeness it should be stated that the early data in the Table are being shown as N (%).  I also found Table 2, very helpful and revealing.

5.        Discussion Page 11. Lines 358 to 374.  I found some of this paragraph a little confusing.  It would be useful to see if any other reviewers felt the same. Like I suspect the authors, I was very surprised that

“Maternal pre-pregnancy BMI was not significantly associated with breastfeeding

intention in contrast to findings from a systematic review that found lower intention to breastfeed among women with obesity.34”

I am not sure I fully understand the authors explanation of this finding in the light of the other literature in this area.  Perhaps the authors could re-read this section and maybe try and simplify the paragraph. However, if other reviewers have not found this a little confusing it might my problem not the authors problem!

6.    Discussion Page 12. Lines 385-387.  I am concerned by the statement.

“Care should be taken generalizing results

          of this study to populations of different demographics, as study participants are primarily

         Hispanic, and live in low-income households in one large urban county in Southern California.”

I think the readers deserved this to be expanded. Exactly how much “care” should be taken? Are the authors suggesting that the data cannot be used to support claims or changes in other populations and that the findings are irrelevant to other populations? If so this is a major weakness and should be stated.

Author Response

Please see the attachment for our responses to the reviewer comments. 

Reviewer 2 Report

Comments and Suggestions for Authors

This great article examines the psychological and sociological aspects in breastfeeding, which are often ill-known and/or heavily overlooked. I have some comments:

Table 1: The prevalence/incidence of breastfeeding in this sample population is rather high in the Western World. Was this study population preselected?

Table 2: The breastfeeding duration is also quite long. Is there a "bias" of better breastfeeding leave after maternity leave in this region?

Author Response

Please see the attachment for our responses to reviewer comments.

Reviewer 3 Report

Comments and Suggestions for Authors

Dear Authors,

Thank you for the opportunity to perform a review of this manuscript.

The manuscript addresses an important issue directly related to the biological condition of society. The duration of the breastfeeding period and its effects are increasingly known. The biological and psychological development of children who are breastfed for a minimum of six months provides a model of the regularity of child development. The results should be applicable to the design of population-based educational programs on the biological and psychological importance of breastfeeding. 

In my opinion, the manuscript presented for review is essentially properly prepared. The theoretical chapters do not raise significant comments from me. Only the use of univariate statistical analyses is questionable. With such extensive data, it is advisable to show which of the analyzed determinants has the greatest impact on the desire to feed the child, taking into account the mutual correlations of the determinants.

The fact of having a life partner, having a steady job, having a lower BMI, etc. - are characteristics highly correlated with each other as well as with the level of education. It would be worthwhile for the authors to use multivariate statistical tests and identify the most important characteristics (after excluding mutual correlations) affecting mothers' declarations of breastfeeding. Such results would have highly relevant application.

In addition, I suggest that the keywords should not overlap with the title - the entire title is keywords.

Kin regards,

reviewer

Author Response

(The authors gave the same response as above.)

Round 2

Reviewer 3 Report

Comments and Suggestions for Authors

Dear Authors,

I am not satisfied with the answer.

The authors did not respond to the comment:

Only the use of univariate statistical analyses is questionable. With such extensive data, it is advisable to show which of the analyzed determinants has the greatest impact on the desire to feed the child, taking into account the mutual correlations of the determinants.

I do not accept the information that this was not the purpose of the article. It is simply that the current form of presentation of the results (univariate analyses) leads to erroneous conclusions.

Once again, I ask you to use multivariate analyses.

Author Response

Dear Authors,

I am not satisfied with the answer.

The authors did not respond to the comment:

Only the use of univariate statistical analyses is questionable. With such extensive data, it is advisable to show which of the analyzed determinants has the greatest impact on the desire to feed the child, taking into account the mutual correlations of the determinants.

I do not accept the information that this was not the purpose of the article. It is simply that the current form of presentation of the results (univariate analyses) leads to erroneous conclusions.

Once again, I ask you to use multivariate analyses.

Response:  We have now performed a multinomial logistic regression analysis for prenatal breastfeeding intention on independent terms for the characteristics with significant bivariate associations with prenatal breastfeeding intention from Table 1. Only survey year, breastfeeding support from WIC, and breastfeeding support from family/friends remained statistically significant. This has now been described in the manuscript (lines 117-131).